# Structural analysis of the *Legionella pneumophila* Dot/Icm type IV secretion system core complex

Clarissa L Durie[1†], Michael J Sheedlo[2†], Jeong Min Chung[1], Brenda G Byrne[3], Min Su[1], Thomas Knight[3], Michele Swanson[3*], D Borden Lacy[2,4,5*], Melanie D Ohi[1*]

[1]Life Sciences Institute, University of Michigan, Ann Arbor, United States; [2]Department of Pathology, Microbiology, and Immunology, Department of Pathology, Vanderbilt University Medical Center, Nashville, United States; [3]Department of Microbiology and Immunology, University of Michigan, Ann Arbor, United States; [4]The Veterans Affairs Tennessee Valley Healthcare System, Nashville, United States; [5]Department of Cell and Developmental Biology, University of Michigan, Ann Arbor, United States

**\*For correspondence:**
mswanson@umich.edu (MS);
borden.lacy@vanderbilt.edu
(DBL);
mohi@umich.edu (MDO)

[†]These authors contributed
equally to this work

**Competing interests:** The
authors declare that no
competing interests exist.

**Reviewing editor:** Andrew P
Carter, MRC Laboratory of
Molecular Biology, United
Kingdom

**Abstract** *Legionella pneumophila* is an opportunistic pathogen that causes the potentially fatal pneumonia Legionnaires' Disease. This infection and subsequent pathology require the Dot/Icm Type IV Secretion System (T4SS) to deliver effector proteins into host cells. Compared to prototypical T4SSs, the Dot/Icm assembly is much larger, containing ~27 different components including a core complex reported to be composed of five proteins: DotC, DotD, DotF, DotG, and DotH. Using single particle cryo-electron microscopy (cryo-EM), we report reconstructions of the core complex of the Dot/Icm T4SS that includes a symmetry mismatch between distinct structural features of the outer membrane cap (OMC) and periplasmic ring (PR). We present models of known core complex proteins, DotC, DotD, and DotH, and two structurally similar proteins within the core complex, DotK and Lpg0657. This analysis reveals the stoichiometry and contact interfaces between the key proteins of the Dot/Icm T4SS core complex and provides a framework for understanding a complex molecular machine.

## Introduction

The Type IV Secretion System (T4SS) is a potent weapon used by some bacteria to infect their host and can deliver effector proteins into eukaryotic cells as well as DNA and/or toxins into bacterial neighbors (*Grohmann et al., 2018*; *Cascales and Christie, 2003*; *Fronzes et al., 2009*). T4SSs are deployed by a variety of human pathogens, such as *Legionella pneumophila, Helicobacter pylori, and Bordetella pertussis* (*Grohmann et al., 2018*; *Cascales and Christie, 2003*; *Fronzes et al., 2009*). In the Gram-negative pathogen *L. pneumophila*, the Dot/Icm T4SS (*Schroeder, 2017*) delivers ~300 effector proteins to the cytoplasm of host cells, in some cases causing Legionnaires' Disease (*Schroeder, 2017*; *Swanson and Hammer, 2000*; *Molofsky and Swanson, 2004*). The T4SSs of Gram-negative bacteria are organized into an inner membrane complex, a core complex that spans the periplasmic space, and in some species an extracellular pilus (*Grohmann et al., 2018*; *Fronzes et al., 2009*; *Waksman, 2019*; *Low et al., 2014*; *Christie et al., 2014*). These T4SSs vary in complexity with some species requiring only 12 components to assemble the complete apparatus. Some systems, such as the Dot/Icm T4SS of *L. pneumophila* and the Cag T4SS of *H. pylori*, are much larger and are constructed from over 20 different proteins, many of which are species-specific (*Chung et al., 2019*; *Purcell and Shuman, 1998*).

The current structural understanding of the large T4SSs has been limited to comparisons to minimized, prototype systems from *Xanthomonas citri* and the pKM101 conjugation system (referred to herein as pKM101). These studies have revealed homologous core structures that are comprised of three components known as VirB7, VirB9 and VirB10 (*Chung et al., 2019*; *Sgro et al., 2018*; *Rivera-Calzada et al., 2013*; *Chandran et al., 2009*). Previous cryo-electron tomography (cryo-ET) studies on the Dot/Icm T4SS have suggested a similar arrangement of some components of this system though no high-resolution data of the intact complex have been obtained to date (*Ghosal et al., 2017*; *Ghosal et al., 2019*; *Chetrit et al., 2018*; *Park et al., 2020*).

## Results and discussion

Towards obtaining a high resolution understanding of the Dot/Icm T4SS we purified from *L. pneumophila* intact core complex particles as evident from negative stain electron microscopy, as previously described (*Kubori and Nagai, 2019*; *Figure 1—figure supplement 1A*). Central to assembly of the apparatus are five proteins that define the core complex: DotC, DotD, DotF, DotG, and DotH (*Ghosal et al., 2017*; *Ghosal et al., 2019*; *Kubori et al., 2014*; *Nagai and Kubori, 2011*). Mass spectrometry analysis of the purification verified the presence of these predicted core components (*Ghosal et al., 2019*; *Kubori and Nagai, 2019*; *Kubori et al., 2014*; *Vincent et al., 2006*), as well as additional proteins identified in *dot* (defect in organelle trafficking) or *icm* (intra-cellular

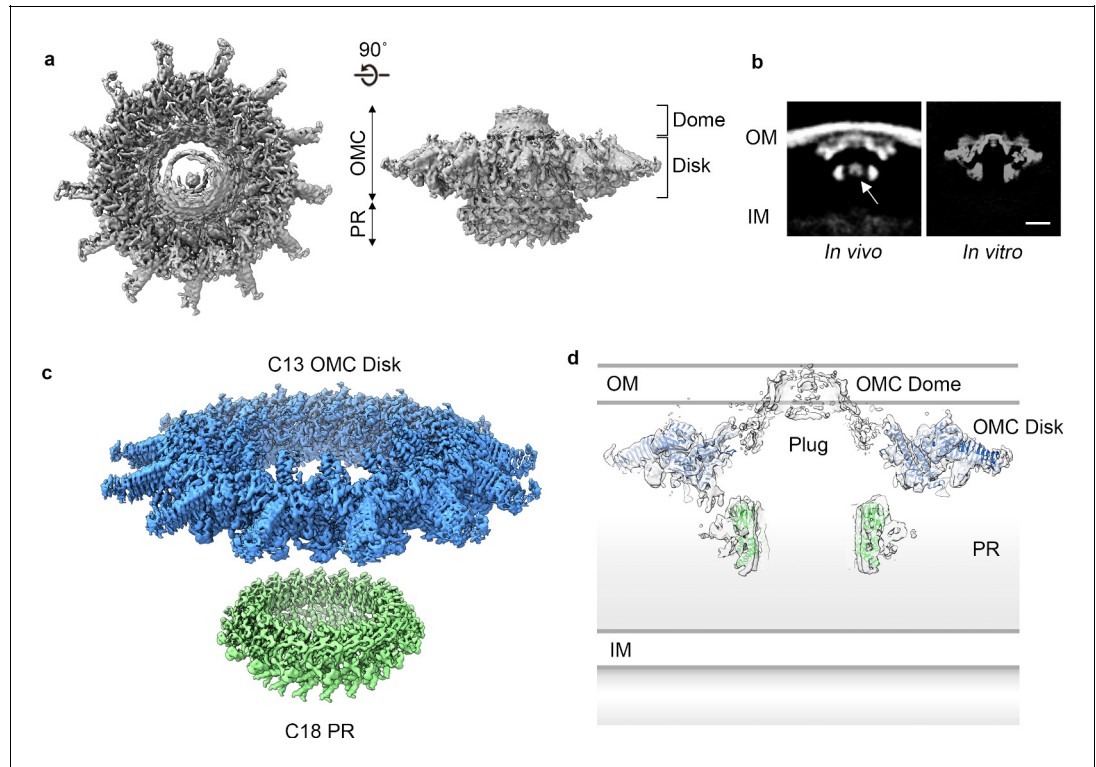

**Figure 1.** Cryo-EM structure of the *L. pneumophila* Dot/Icm T4SS. (**a**) Reconstruction of the *L. pneumophila* Dot/Icm T4SS particles at 4.6 Å with no symmetry applied reveals two parts, the outer membrane cap (OMC), composed of a central dome and flat disk, and a periplasmic ring (PR). (**b**) Comparison of the central sections through the longitudinal plane of the T4SS 3D density determined by cryo-ET of intact *L. pneumophila* (left panel, EMD 0566) (*Ghosal et al., 2019*) with arrow indicating plug density or by cryo-EM of purified particles (this study) (right panel). OM, Outer Membrane, IM, Inner Membrane, Scale bar 10 nm. (**c**) Combined high resolution structures of the *L. pneumophila* Dot/Icm T4SS that include the 3.5 Å OMC disk (blue) with 13-fold symmetry and the 3.7 Å PR (green) with 18-fold symmetry. (**d**) Central axial slice view showing how atomic models of the OMC disk (blue) and PR (green) fit into the C1 3D map of the Dot/Icm T4SS (light gray).

The online version of this article includes the following figure supplement(s) for figure 1:

**Figure supplement 1.** Asymmetric reconstruction of the Dot/Icm T4SS.

**Figure supplement 2.** Flow chart of cryo-EM processing steps.

**Figure supplement 3.** Symmetric reconstructions of the OMC and PR of the Dot/Icm T4SS.

**Table 1.** Dot proteins present in isolated complex sample isolated from wild type strain (Lp02).

| Identified proteins | Gene Number* | Spectral Counts[†] | | |
| --- | --- | --- | --- | --- |
| | | Prep 1 | Prep 2 | Prep 3 |
| DotG[‡] | Q5ZYC1 | 112 | 114 | 195 |
| **DotF** | Q5ZYC0 | 94 | 69 | 101 |
| DotA | Q5ZS33 | 38 | 65 | 60 |
| DotO | Q5ZYB6 | 37 | 38 | 47 |
| **DotH** | Q5ZYC2 | 28 | 19 | 28 |
| IcmF | Q5ZYB4 | 15 | 18 | 36 |
| IcmX | Q5ZS30 | 19 | 13 | 28 |
| DotL | Q5ZYC6 | 10 | 26 | 20 |
| **DotC** | Q5ZS44 | 9 | 11 | 16 |
| **DotD** | Q5ZS45 | 11 | 9 | 14 |
| DotB | Q5ZS43 | 16 | 11 | 7 |
| **Lpg0657** | Q5ZXS4 | 6 | 4 | 16 |
| DotM | Q5ZYC7 | 2 | 14 | 3 |
| **DotK** | Q5ZYC5 | 2 | 6 | 10 |
| IcmW | Q5ZS31 | 5 | 7 | 5 |
| DotN | Q5ZYB7 | 2 | 6 | 4 |
| DotI | Q5ZYC3 | 1 | 3 | 5 |
| IcmS | Q5ZYD0 | | 4 | 1 |
| IcmT | Q5ZYD1 | 1 | 1 | 3 |
| IcmV | Q5ZS32 | 1 | 1 | 2 |

*UniProtKB Accession Number.

[†]Proteins were identified by searching the MS/MS data against *L. pneumophila* (UniProt; 2930 entries) using Proteome Discoverer (v2.1, Thermo Scientific). Search parameters included MS1 mass tolerance of 10 ppm and fragment tolerance of 0.1 Da. False discovery rate (FDR) was determined using Percolator and proteins/peptides with a FDR of ≤1% were retained for further analysis. Complete results are in **Supplementary file 1** in supplementary material.

[‡]Predicted core T4SS components and additional components identified in this structure are in bold.

multiplication) genetic screens (*Segal et al., 1998*; *Segal and Shuman, 1999*; *Vogel et al., 1998*; *Table 1*). We vitrified this sample and, although particles adopt a preferred orientation in vitrified ice, both *en face* and side views are observed, allowing for 3D reconstruction (*Figure 1A*, *Figure 1—figure supplement 1B,C*, and *Figure 1—figure supplement 2*). The Dot/Icm T4SS is ~400 Å wide and ~165 Å long, consistent in shape and size with T4SS complexes visualized in intact *L. pneumophila* using cryo-ET (*Ghosal et al., 2017*; *Ghosal et al., 2019*; *Chetrit et al., 2018*; *Park et al., 2020*; *Figure 1B*). The global resolution of the map without imposed symmetry is 4.6 Å, with the highest resolution regions near its center (*Figure 1—figure supplement 1D,E*). The map can be divided into two major regions: an outer membrane cap (OMC) and a hollow periplasmic ring (PR). The OMC can be further subdivided into two features, a central dome and a flat disk containing 13 arms that extend radially outward (*Figure 1A,C,D*). While cryo-ET analysis of the Dot/Icm T4SS in intact cells included a stalk bridging the PR and the inner membrane (*Ghosal et al., 2019*), this portion of the complex is not observed in the reconstruction of the purified T4SS, likely due to dissociation during purification. An axial section through the map in *Figure 1B,D* reveals a large cavity running through the T4SS, starting from the bottom of the PR and extending to the OMC region that spans the outer membrane, although there appears to be density in the central cavity closest to the outer membrane. This is perhaps the 'plug' seen in the cryo-ET analysis of *in situ* T4SS (*Ghosal et al., 2019*).

The dome of the T4SS OMC is positioned within the center of the map and is about ~50 Å high and ~100 Å wide. Attempts to refine the dome by imposing different symmetries did not improve

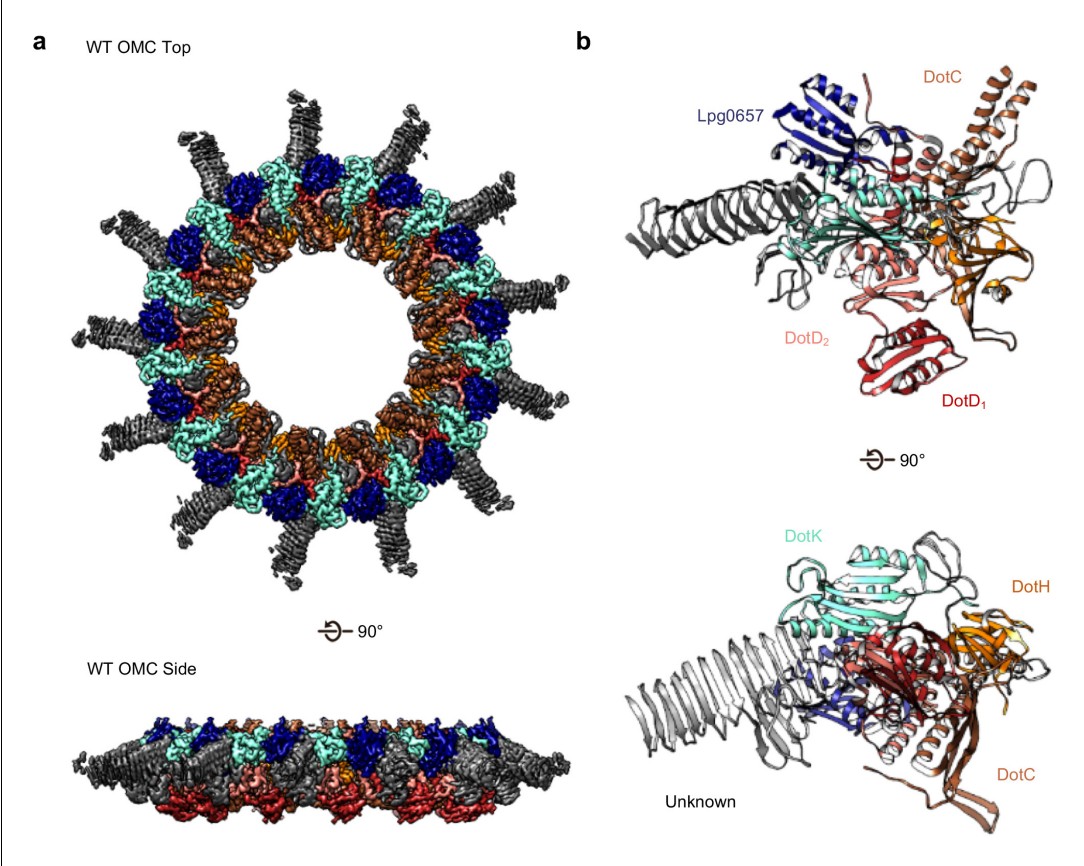

**Figure 2.** The OMC disk of the Dot/Icm T4SS. (**a**) The OMC disk of the Dot/Icm T4SS was reconstructed from samples that were purified from the WT strain (**b**) We have defined an asymmetric unit that is comprised of DotC (brown), DotD$_1$ (red), DotD$_2$ (salmon), DotH (orange), DotK (cyan), Lpg0657 (blue), and three unknown chains (gray).

The online version of this article includes the following figure supplement(s) for figure 2:

**Figure supplement 1.** Correlation between the OMC cryo-EM map and models.
**Figure supplement 2.** Model-map correlation for each protein within the OMC.

the resolution; therefore no clear symmetry was defined. In other T4SSs that have been structurally characterized, the dome is a contiguous part of the OMC, shares the same symmetry, and is clearly composed of organized α-helices (*Chung et al., 2019*; *Sgro et al., 2018*; *Chandran et al., 2009*). While we do not see individual helices in our map, in the C1 reconstruction the narrow opening of the *L. pneumophila* dome is ~40 Å in diameter, a dimension within the range of pore sizes observed in the OMC of other species (*Figure 1A*; *Chung et al., 2019*; *Sgro et al., 2018*; *Chandran et al., 2009*).

Using symmetry and focused refinement, we determined a 3.5 Å resolution map of the OMC disk and a 3.7 Å map of the PR (*Figure 1C*, *Figure 1—figure supplement 2*, and *Figure 1—figure supplement 3*). Notably, while the disk exhibits the expected 13-fold symmetry observed previously (*Ghosal et al., 2017*; *Ghosal et al., 2019*; *Chetrit et al., 2018*; *Park et al., 2020*; *Hu et al., 2019*), the PR contains 18-fold symmetry (*Figure 1C* and *Figure 1—figure supplement 3A,D*). While the possibility of this symmetry mismatch was postulated from low-resolution *in situ* structures of the Dot/Icm T4SS (*Park et al., 2020*), the symmetry of the different regions was not determined. Interestingly, a similar symmetry mismatch occurs between the *H. pylori* T4SS OMC and PR: its OMC contains 14-fold symmetry while the PR has 17-fold symmetry (*Chung et al., 2019*). The resolution of the Dot/Icm T4SS OMC disk and PR maps made it possible to construct models of the proteins in these regions (*Figure 1C,D*).

The OMC disk makes up the pinwheel-shaped portion of the T4SS and is organized into a thick central region with 13 arms extending radially outward (*Figure 2A*). The disk is ~75 Å along the axial

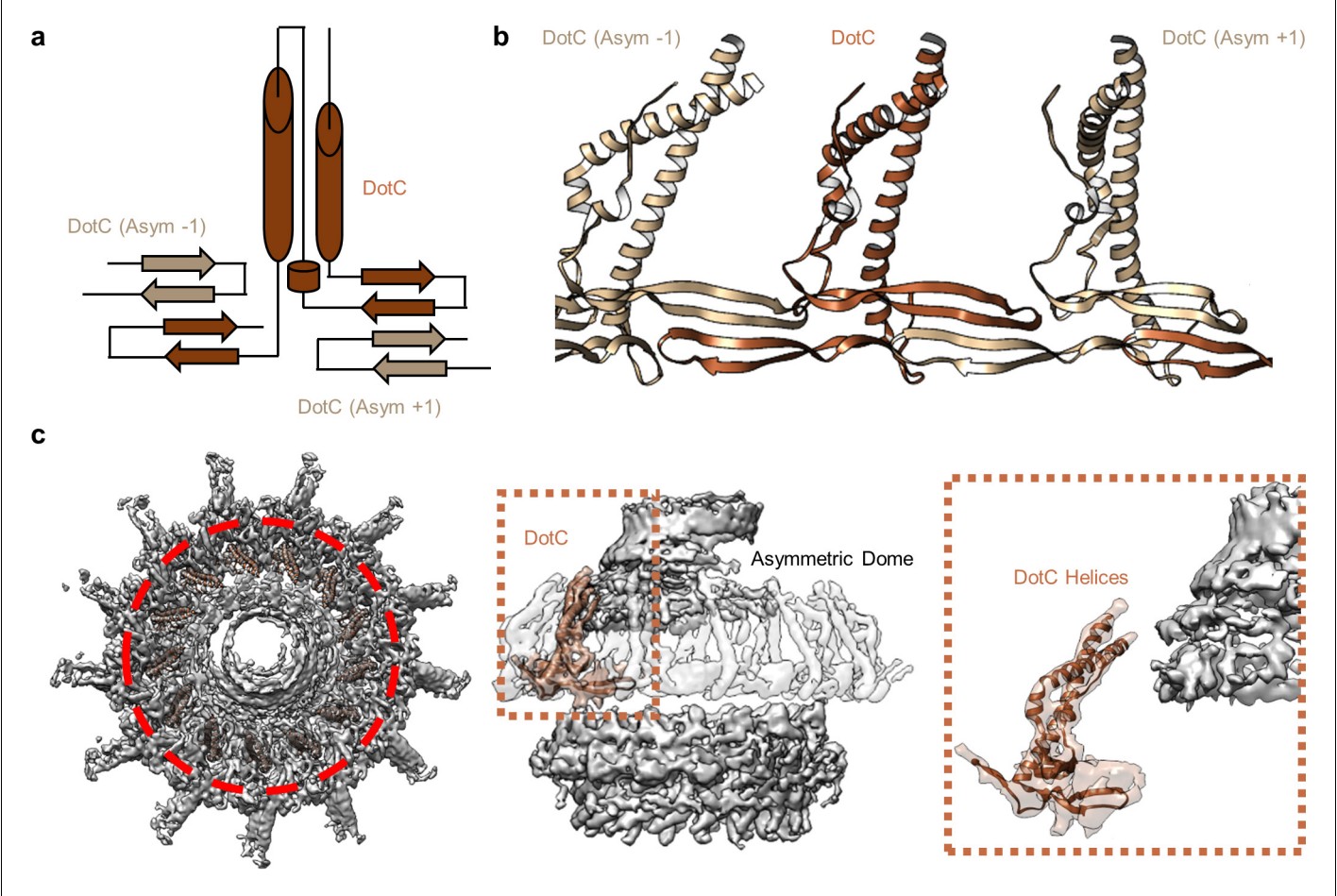

**Figure 3.** DotC forms a nearly uninterrupted β-sheet about the center of the complex. (**a**) DotC is composed of two large α-helices and two β-sheets. (**b**) These sheets are formed between asymmetric units and comprise a predominantly electrostatic structure that lines the central chamber. (**c**) DotC is adjacent to the poorly resolved portion of the C1 map (circled with a red dotted line, left). The positioning of DotC is such that a gap exists between it and the asymmetric portion of the map (brown dotted line, center). This results in a small predicted interface between DotC and the central dome (right).

dimension with an interior chamber ~150 Å wide. The disk is also thin compared to other structurally characterized T4SS OMCs and contains no distinct inner or outer layers (*Chung et al., 2019*; *Sgro et al., 2018*; *Chandran et al., 2009*). Within the disk, we unambiguously traced and identified DotC, DotD, DotK and DotH along with the protein Lpg0657 (*Goodwin et al., 2016*; *Figure 2B*, *Figure 2—figure supplements 1* and *2*). Of these, only structures of the C-terminal domain of DotD (PDB 3ADY) and the structure of Lpg0657 (PDB 3LDT) had been previously reported (*Nakano et al., 2010*). Notably, rather than an equimolar ratio, the components of the OMC exist at a ratio of 2:1:1:1:1 (DotD:DotC:DotH:DotK:Lpg0657) (*Figure 2B*).

At the center of the OMC is an elongated fold that is comprised of α-helices and β-strands that we have identified as DotC (residues 58–161 and 173–268). DotC is folded such that two large α-helices protrude toward the outer membrane and are flanked on either side by two β-strands (*Figure 3A*). The two β-strands adjacent to the central α-helices fold into a nearly uninterrupted β-sheet that is formed between asymmetric units and consists of a generally hydrophilic surface that runs about the central cavity (*Figure 3B*). When docked into the asymmetric reconstruction, this β-sheet lines the poorly resolved central section of the map. From these data, the only clear contact that is made between DotC and the central pore is a small interface at the top of the two long α-helices, which may explain why this portion is not well resolved in the maps (*Figure 3C*). A search of the protein data bank yielded a wide array of potential structural homologs, including a number of

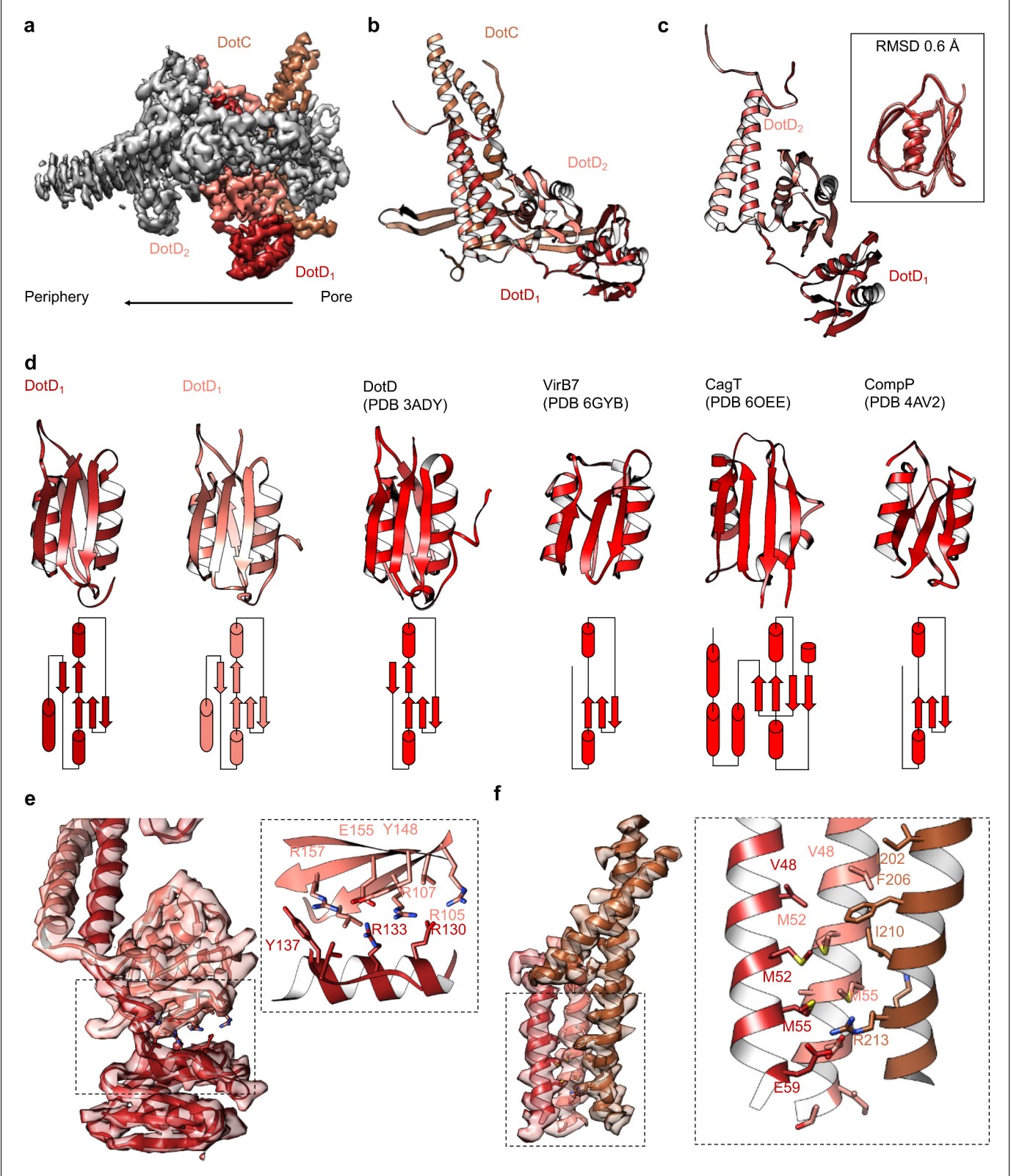

**Figure 4.** Components in the center of the Dot/Icm T4SS. (**a–b**) On the peripheral side, DotC is observed with two copies of DotD, shown in red and salmon. (**c**) The two copies of DotD vary little in their overall organization, deviating by an RMSD of only 0.6 Å within the C-terminal domain as shown in the inset. (**d**) Both copies of DotD observed within this map adopt a fold that is similar to the previously reported crystal structure as well as related VirB7 homologs. (**e**) The interface that is formed between DotD1 and DotD2 is formed by a number of electrostatic and polar interactions as shown in

*Figure 4 continued on next page*

*Figure 4 continued*

the inset. (**f**) The N-terminal α-helices of DotD$_1$ and DotD$_2$ form the bulk of the interface observed with DotC, made up of a hydrophobic interface as shown in the inset.

channels and transporters; most of these share little homology to DotC overall (*Holm, 2019*). On the peripheral side, DotC makes contact with two copies of DotD, which we have called DotD$_1$ and DotD$_2$ (*Figure 4A–C*). The core folds of these proteins are similar to that of other components of large bacterial complexes such as VirB7 homologs from other T4SSs as well as components of type four pilus systems (*Figure 4D*). The interface between the two copies of DotD is mediated by electrostatic interactions and hydrogen bonds (*Figure 4E*). The N-terminus, which was not fully visualized in the previously reported DotD crystal structure (*Nakano et al., 2010*), is α-helical and extends from the middle of the disk toward the pore, forming a dimer that interacts with the central α-helices of DotC (*Figure 4F*).

Adjacent to the N-terminus of DotD we have modeled a small globular fold that we have identified as DotK. DotK is positioned adjacent to the outer membrane, consistent with previous studies (*Figure 5A,B*; *Ghosal et al., 2019*). By chain tracing, we identified a second, similar fold near DotK. After docking the structure of DotK into the density, we noted that although the model fits well globally, some portions of the model were not supported by the density locally. Therefore, we initiated a DALI search for structurally similar molecules that identified Lpg0657 (PDB 3LDT) as a structural homolog and candidate for this density (*Holm, 2019*; *Figure 5A–D*). Though Lpg0657 has not been shown to directly interact with the Dot/Icm T4SS in prior studies, it is vital for *L. pneumophila* replication *in vitro* (*Goodwin et al., 2016*). In fact, Goodwin and colleagues hypothesized that Lpg0657 might interact with the Dot/Icm T4SS (*Goodwin et al., 2016*). Upon refining this model into the density, we noted that all features of Lpg0657 fit well into the density both globally and locally (*Figure 5E*). The presence of Lpg0657 is corroborated by mass spectrometry data of this sample (*Table 1*). The structures of both DotK and Lpg0657 resemble peptidoglycan binding domains (*Lin et al., 2019*); however, density corresponding to peptidoglycan was not observed within the binding cleft in either DotK or Lpg0657, and several key residues known to mediate peptidoglycan interactions are not present. Thus, we suspect that neither protein mediates direct interaction with peptidoglycan. Notably, although the two proteins share a similar fold, they make contact with different members of the T4SS. DotK contacts the C-terminal domain of DotD$_2$, whereas Lpg0657 interacts with the N-terminal α-helices of both DotD$_1$ and DotD$_2$. This architecture is likely due to differences in the primary sequences of DotK and Lpg0657 that dictate the arrangement of these two similar proteins within the apparatus.

On the periplasmic side of the OMC is another small globular fold that we identified as the C-terminal domain of DotH. DotH consists of two β-sheets that are arranged in a β-sandwich fold (*Figures 2B* and *6A*). Our subsequent structural search identified the VirB9 homolog TraO as the protein with the highest degree of structural similarity to DotH (PDB 3JQO, *Figure 6B*). Indeed, TraO harbors a conserved fold that is also observed in similar proteins such as VirB9 and CagX (*Figure 6C*). DotH could not have been predicted as a VirB9 homolog based on its primary structure, as very little conservation is observed between the two proteins (*Figure 6D*). Similar to its counterparts in other species (*Chung et al., 2019*; *Sgro et al., 2018*; *Rivera-Calzada et al., 2013*; *Chandran et al., 2009*), the C-terminal domain of DotH begins with an α-helix positioned near the center of the map which extends outward from the periplasm (*Figure 6E*).

Within the OMC disk we traced three poly-alanine chains that could not be unambiguously identified as any component of the Dot/Icm T4SS (*Figures 2B* and *7A*). The first (chain 1) is a two-lobed polypeptide where ~70 residues form a series of loops (*Figure 7B*). This unknown protein is positioned atop the OMC making contact with DotC, DotD, DotH, and presumably the outer membrane (*Figure 7A*). Located on the periphery of the map and extending outward radially, we have modeled a 22-strand β-helix (chain 2) (*Figure 7C*). Bound to this β-helix is a small fold consisting of six strands (chain 3). It is currently unclear if this domain represents an insertion in the β-helix or a distinct protein (*Figure 7D*). Based on previous reports it is tempting to speculate that the β-helix structure may correspond to DotG (*Ghosal et al., 2019*). However, a register could not be identified in this part of the density due to low resolution, and, additionally, tomography studies have suggested the pentapeptide repeats reside in the stalk of the Dot/Icm T4SS (*Ghosal et al., 2019*).

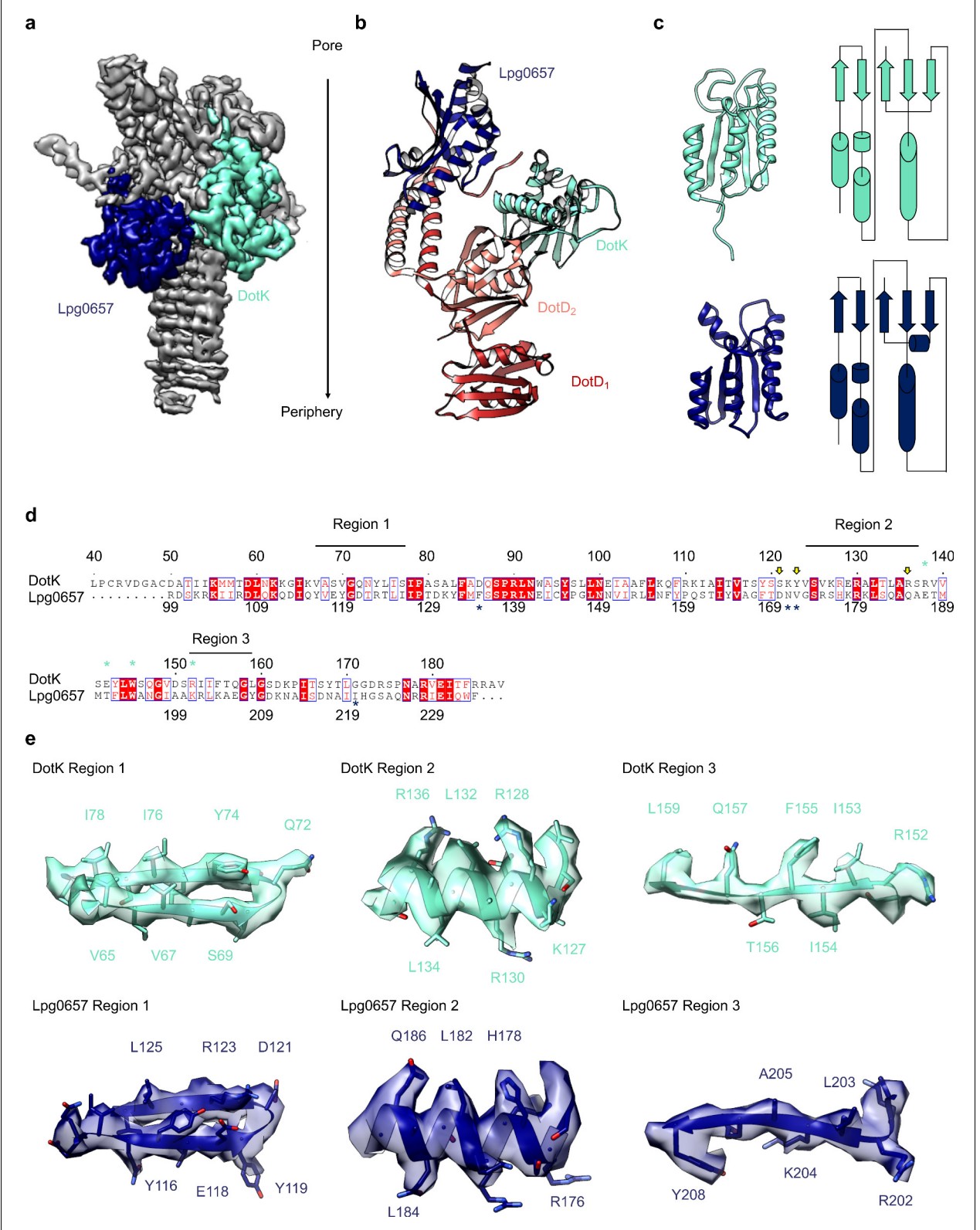

**Figure 5.** Components of the Dot/Icm T4SS near the outer membrane. (**a**) Two nearly identical folds were discovered on the outer membrane side of the OMC that correspond to DotK (cyan) and Lpg0657 (blue). (**b**) DotK and Lpg0657 arrange such that each contacts both copies of DotD within the asymmetric unit, although they use different surfaces to mediate this interaction. (**c**) The two folds have a nearly identical topology. Proteins colored as
*Figure 5 continued on next page*

*Figure 5 continued*

in panel (a). (d) Although these folds are very similar, at least three regions could be used to distinguish between the two proteins in this map. (e) Characteristic density is observed for each protein and supports the assignment of each protein within the map.

To test whether DotG may be localized to the dome region, the radial arms, or both, we isolated and structurally characterized the T4SS in a *Lp* mutant lacking DotG (Δ*dotG*) (*Figure 8A*, *Figure 8—figure supplements 1*, *2* and *3*). Although Δ*dotG* mutant bacteria assemble a Dot/Icm T4SS, they are defective for secretion and replication in host cells (*Vogel et al., 1998*). Mass spectrometry from this mutant purification confirms DotG is absent (*Table 2*). The ΔDotG T4SS also lacks DotF (*Table 2*). Since Western blotting analysis of purified complexes from a Δ*dotG* mutant bacteria showed wild-type levels of DotF (*Kubori et al., 2014*), the *dotG* deletion-insertion allele analyzed here may be polar on expression of the downstream *dotF* gene. Our structural analysis of ΔDotG T4SS shows that while it contains the OMC disk and the 13 extended arms, the complex lacks both the dome and the

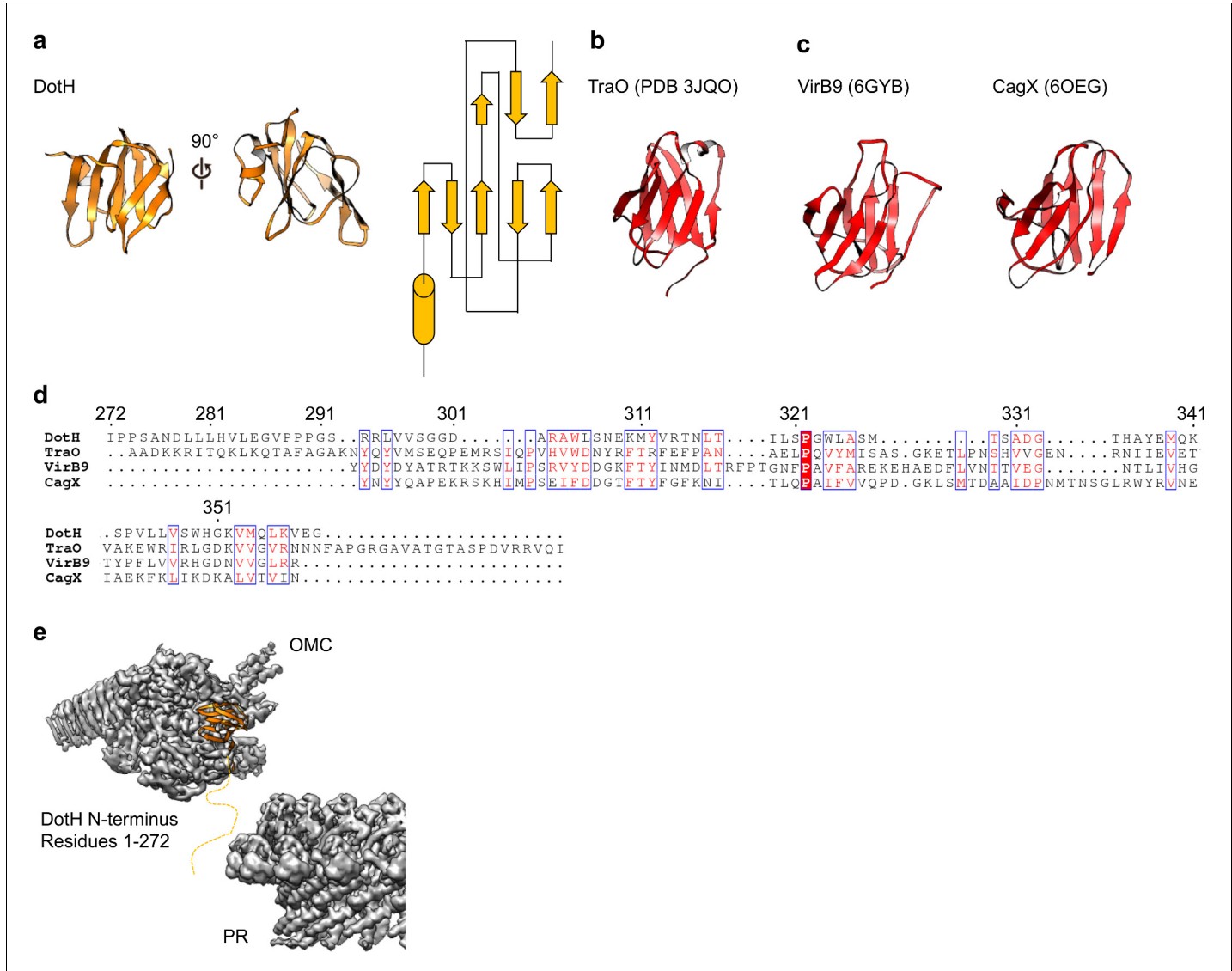

**Figure 6.** DotH is a VirB9 homolog and positioned in the center of the map. (a) The C-terminal domain of DotH was discovered within the center of the OMC map and consists of a β-sandwich fold. (b) The structure is similar to other T4SS proteins such as the C-termini of TraO, VirB9, and CagX. (c) Though the structures are similar, little sequence similarity (d) is observed throughout the family. (e) DotH is positioned such that the N-terminus (consisting of 272 residues) is positioned between the OMC and PR.

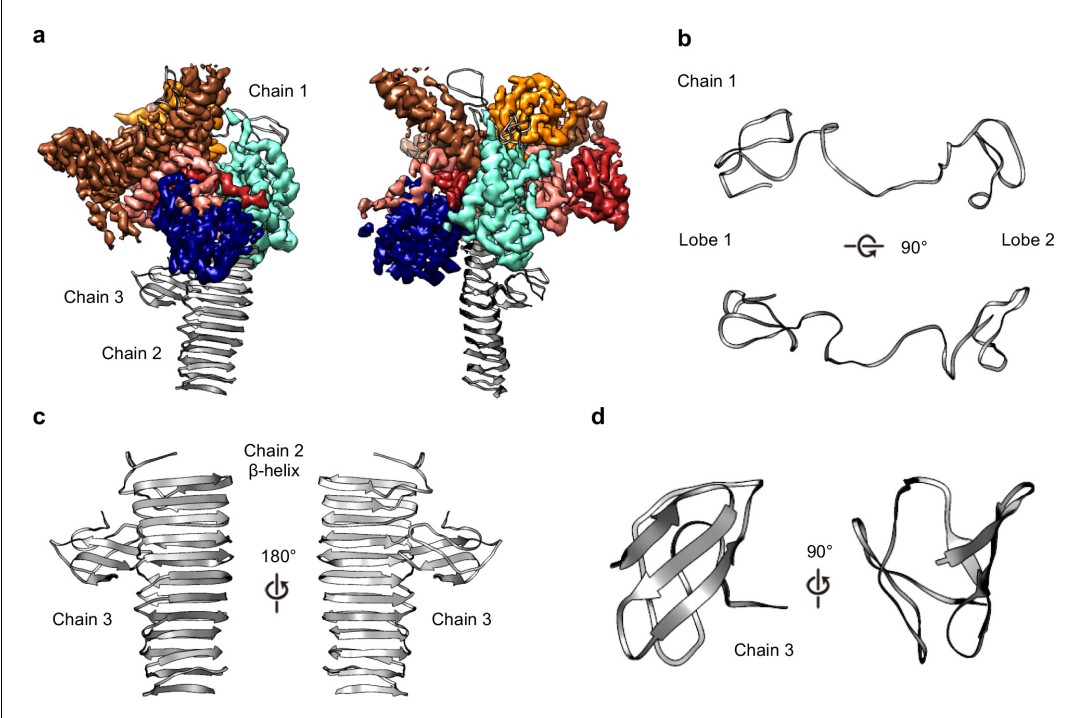

**Figure 7.** Three unknown components were modeled as polyalanine chains. (**a**) Three chains were constructed as polyalanine models, denoted as chain 1–3 and shown in gray. (**b**) Chain 1 is observed near the outer membrane and consists of loops forming two lobes. (**c**) Chain 2 consists of a long β-helix that interacts with chain 3. (**d**) Chain 3 is a small globular fold that is comprised of only β-strands.

PR (*Figure 8A* and *Figure 8—figure supplement 1D*). This finding is in agreement with the proposed model for the overall organization of both ΔDotG T4SS and ΔDotFΔDotG T4SS complexes predicted from immunoblot analysis and images of negatively stained T4SS complexes lacking either DotF or DotG (*Kubori et al., 2014*). All components modeled in the OMC from the wild-type T4SS were also present within the ΔDotG T4SS complexes, supporting the identifications described above (*Table 2*, *Figure 8B*). In other T4SSs, the dome region of the complex is comprised of homologous proteins known as VirB10 (*X. citri*), CagY (*H. pylori*), or TraF (pKM101). By sequence homology, the C-terminus of DotG is predicted to be structurally similar to these components (*Figure 8—figure supplement 4A,C*; *Chung et al., 2019*; *Sgro et al., 2018*; *Rivera-Calzada et al., 2013*; *Chandran et al., 2009*; *Ghosal et al., 2017*; *Nagai and Kubori, 2011*; *Hu et al., 2019*). Thus, we propose that the C-terminus of DotG makes up the dome of the *Lp* T4SS (*Figure 8—figure supplement 4B*). In agreement with this, we note that a model of the C-terminus of DotG based on the structure of CagY (generated in Swiss Model) fits into the dome density (*Figure 8—figure supplement 4B*), though its identity as DotG needs to be confirmed (*Waterhouse et al., 2018*). The rest of DotG and DotF may contribute to the structural interface between the OMC and the PR and/or form a portion of the structure of the PR.

Our predicted placement of DotF and DotG is consistent with two previous reports (*Ghosal et al., 2019*; *Vincent et al., 2006*). Biochemical studies showed that DotG associates closely with DotH and DotC, shown here to form part of the OMC (*Figure 2B*), and that both DotF and DotG are integral inner membrane proteins that also associate with the outer membrane (*Vincent et al., 2006*). A cryo-ET analysis of the T4SS in a Δ*dotG* strain reported missing density from the stalk, plug, and dome compared to subtomogram averages from T4SS complexes in a wild-type strain (*Ghosal et al., 2019*). Moreover, these cryoET studies showed that the T4SS subtomogram averages in a Δ*dotF* strain lack density in the periplasmic region compared to complex in a wild-type strain (*Ghosal et al., 2019*).

The PR has been observed in the recently characterized single particle cryo-EM reconstruction of the *H. pylori* Cag T4SS and tomography studies of both *H. pylori* and *L. pneumophila* T4SSs

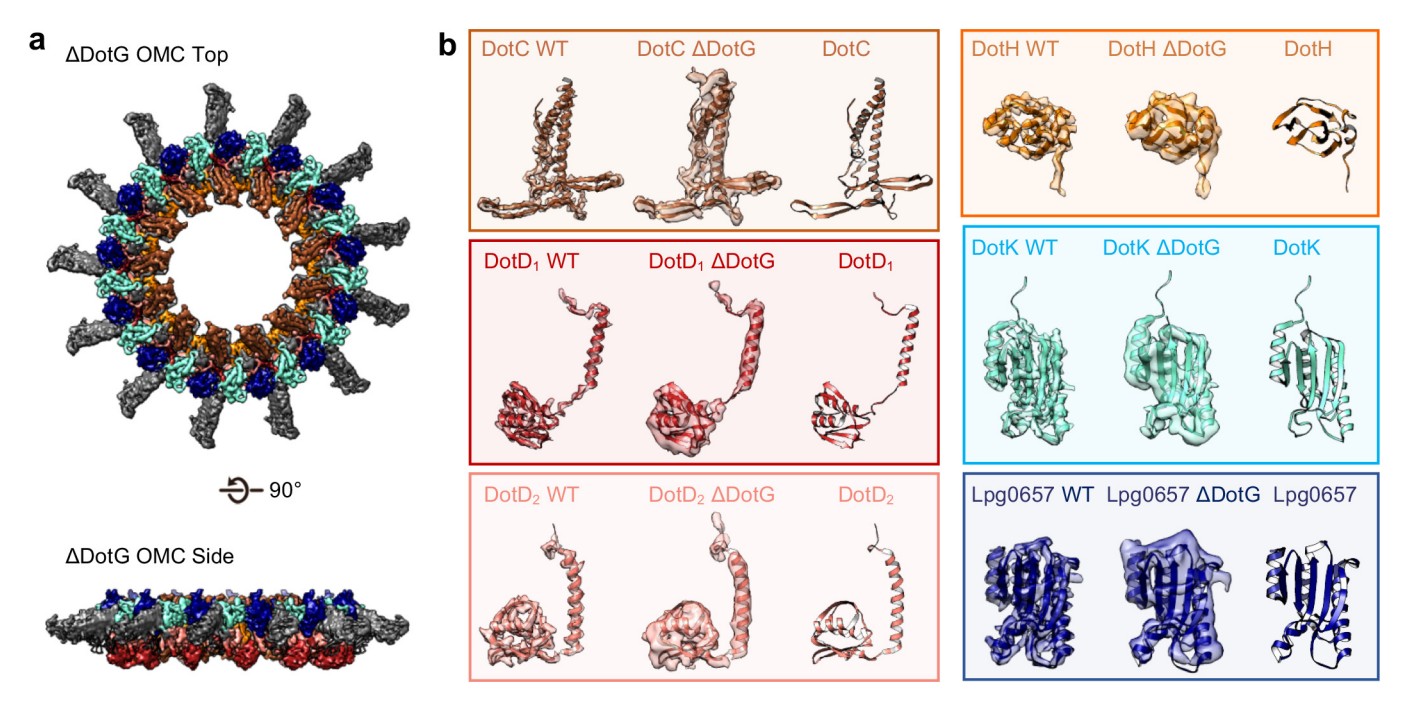

**Figure 8.** The OMC disk of the ΔDotG T4SS. (a) The OMC disk of the ΔDotG T4SS was reconstructed from samples that were purified from a strain lacking DotG (Δ*dotG*). (b) All proteins within the asymmetric unit adopt nearly identical orientations in complexes identified from the WT and mutant strain. (Colored as in panel a).

The online version of this article includes the following figure supplement(s) for figure 8:

**Figure supplement 1.** 3D reconstruction of the T4SS purified from a Δ*dotG* mutant.
**Figure supplement 2.** Correlation between the *Lp* ΔDotG OMC cryo-EM map and models.
**Figure supplement 3.** Model-map correlation for each protein within the *Lp* ΔDotG OMC.
**Figure supplement 4.** Homology model of proposed dome component DotG.

(*Chung et al., 2019*; *Ghosal et al., 2017*; *Ghosal et al., 2019*; *Chetrit et al., 2018*; *Park et al., 2020*; *Hu et al., 2019*; *Chang et al., 2018*). The resolution in this region of our Dot/Icm T4SS map was sufficient to model two distinct polyalanine chains within the PR (*Figure 9A,B*). The backbone trace of one of these chains revealed a structure homologous to the N-terminus of *X. citri* VirB9 and similar to a polyalanine model of the PR constructed from the *H. pylori* T4SS (*Figure 9C*). The other, as of yet unidentified, density within the PR is comprised of a single α-helix followed by an extended loop that spans the entire length of the PR. Although the identity of either protein is currently not clear, prime candidates are either DotG or DotF, two proteins present in our preparations but not confidently localized in our maps (*Table 1*, *Figure 2B*). In addition, the entire PR is missing from the ΔDotG T4SS (*Table 2*, *Figure 7A*). As was reported previously for the *H. pylori* Cag T4SS (*Chung et al., 2019*), while the *Lp* OMC and PR make physical contact in the lower resolution map with no applied symmetry (*Figure 9D*), the connections are lost in the refined structures due to the symmetry mismatch.

The high-resolution structure of the *L. pneumophila* Dot/Icm T4SS allows us to compare the structural organization shared between the *H. pylori* Cag T4SS and the *L. pneumophila* Dot/Icm T4SS (*Figure 10*). Both structures display different symmetry within the OMC and PR (*Chung et al., 2019*); however, the symmetry mismatch itself is conserved, suggesting that this feature is important to the function of secretion systems. However, many questions remain about which T4SS components are most important for translocation efficiency, how substrates are recognized, and how effectors are engaged. To address these remaining questions in the context of the Dot/Icm T4SS, future studies need to characterize the presently unidentified chains and determine the roles and locations of DotF and DotG in this complex. As our high-resolution structure of the OMC revealed unexpected core components (DotK and Lpg0657), additional insights into the stalk and coupling

**Table 2.** Dot proteins present in isolated complex sample isolated from Δ*dotG* deletion strain.

| Identified proteins | Gene Number* | Spectral Counts† | | |
|---|---|---|---|---|
| | | Prep 1 | Prep 2 | Prep 3 |
| IcmF | Q5ZYB4 | 28 | 85 | 72 |
| DotA | Q5ZS33 | 20 | 94 | 63 |
| IcmX | Q5ZS30 | 15 | 88 | 25 |
| **DotH**‡ | Q5ZYC2 | 11 | 81 | 25 |
| **DotC** | Q5ZS44 | 12 | 58 | 17 |
| DotO | Q5ZYB6 | 16 | 27 | 38 |
| **Lpg0657** | Q5ZXS4 | 7 | 43 | 13 |
| DotL | Q5ZYC6 | 8 | 15 | 32 |
| DotB | Q5ZS43 | 10 | 23 | 13 |
| **DotK** | Q5ZYC5 | 6 | 26 | 6 |
| **DotD** | Q5ZS45 | 3 | 27 | 6 |
| IcmW | Q5ZS31 | 5 | 9 | 9 |
| DotI | Q5ZYC3 | 1 | 10 | 4 |
| IcmV | Q5ZS32 | 2 | 1 | 11 |
| DotU | Q5ZYB3 | 1 | 5 | 2 |
| DotM | Q5ZYC7 | 0 | 2 | 6 |
| IcmS | Q5ZYD0 | 2 | 2 | 4 |

*UniProtKB Accession Number.

†Proteins were identified by searching the MS/MS data against *L. pneumophila* (UniProt; 2930 entries) using Proteome Discoverer (v2.1, Thermo Scientific). Search parameters included MS1 mass tolerance of 10 ppm and fragment tolerance of 0.1 Da. False discovery rate (FDR) was determined using Percolator and proteins/peptides with a FDR of ≤1% were retained for further analysis. The complete results are shown in **Supplementary file 2** in the supplemental material.

‡Predicted core T4SS components and additional components identified in this structure are in bold.

proteins of the Dot/Icm T4SS are also needed. This first high-resolution structure of the Dot/Icm T4SS shows the importance of complex intermolecular interactions between core components to build a large OMC and highlights the conservation of symmetry mismatch in complex T4SSs, suggesting that both structural features are important for the function of these very large transport systems.

## Materials and methods

### Key resources table

| Reagent type (species) or resource | Designation | Source or reference | Identifiers | Additional information |
|---|---|---|---|---|
| Strain, strain background (*Legionella pneumophila*) | Lp02; WT | PMID:23717549 | | |
| Strain, strain background (*Legionella pneumophila*) | Δ*dotG* | This paper | | Δ*dotG* deletion-insertion mutant |
| Recombinant DNA reagent | pKD3 (plasmid) | Sigma Aldrich | RRID:AddGene_45604 | template plasmids for frt-flanked cat cassette |
| Sequence-based reagent | dotG-F | This paper | PCR primers | aaagcactccacctaagcctacag |
| Sequence-based reagent | dotG-R | This paper | PCR primers | aaaaattagccaagcccgacctg |

*Continued on next page*

*Continued*

| Reagent type (species) or resource | Designation | Source or reference | Identifiers | Additional information |
|---|---|---|---|---|
| Sequence-based reagent | dotG-P0 | This paper | PCR primers | aaatcatgcaactcaaggtagaagggttataagcaaatgtgtgtaggctggagctgcttc |
| Sequence-based reagent | dotG-P2 | This paper | PCR primers | tatccgccatcaaattaaattgttgtaacatcctggcatatgaatatcctccttagttcc |
| Commercial assay or kit | ProteoSilver Plus Silver Stain Kit | Sigma Aldrich | PROTSIL2-1KT | |
| Software, algorithm | Leginon | PMID:15890530 | | |
| Software, algorithm | MotionCor2 | PMID:28250466 | | |
| Software, algorithm | CTFFind4 | PMID:26278980 | | |
| Software, algorithm | cryoSPARC | PMID:28165473 | | |
| Software, algorithm | RELION | PMID:27685097 PMID:30412051 | | |
| Software, algorithm | Coot | PMID:20383002 | | |
| Software, algorithm | UCSF Chimera | PMID:15264254 PMID:29340616 | | |
| Software, algorithm | PHENIX | PMID:29872004 | | |
| Software, algorithm | DALI server | PMID:31263867 | | |

## Preparation of strains

*L. pneumophila* was cultured in ACES (Sigma)-buffered yeast extract broth at pH 6.9 supplemented with 0.1 mg/ml thymidine, 0.4 mg/ml L-cysteine, and 0.135 mg/ml ferric nitrate or on solid medium of this broth supplemented with 15 g/liter agar and 2 g/liter charcoal. The *L. pneumophila* laboratory strain Lp02, a thymidine auxotroph derived from the clinical isolate Philadelphia-1 (*Rao et al., 2013*), was utilized as the wild-type strain. The *dotG* locus of Lp02 was replaced with a *cat* cassette encoding chloramphenicol resistance by homologous recombination as previously described (*Bryan et al., 2013*). The wild-type and Δ*dotG* alleles were amplified using primers dotG-F (5'-aaagcactccacctaagcctacag-3') and dotG-R (5'-aaaaattagccaagcccgacctg-3'). The *cat* cassette was amplified from plasmid pKD3 (*Datsenko and Wanner, 2000*) using primers dotG-P0 (5' aaatcatgcaactcaaggtagaagggttataagcaaatgtgtgtaggctggagctgcttc-3') and dotG-P2 (5'-tatccgccatcaaattaaattgttgtaacatcctggcatatgaatatcctccttagttcc-3'). The Δ*dotG* deletion-insertion mutant was selected and purified on medium supplemented with 5 μg/ml chloramphenicol.

## Complex isolation

Complexes were isolated from wild-type *L. pneumophila* strain Lp02 and the Δ*dotG* mutant strain as described (*Kubori and Nagai, 2019*; *Kubori et al., 2014*). Cells were suspended in 140 mL of buffer containing 150 mM Trizma base pH 8.0, 500 mM NaCl, and EDTA-free Complete protease inhibitor (Roche) at 4°C. The suspension was incubated on the benchtop, with stirring, until it reached ambient temperature. PMSF (final concentration 1 mM), EDTA (final concentration 1 mM), and lysozyme (final concentration 0.1 mg/mL) were added and the suspension was incubated at ambient temperature for an additional 30 min. Bacterial membranes were lysed using detergent and alkaline lysis. Triton X-100 (20% w/v) with AG501-X8 resin (BioRad) was added dropwise, followed by MgSO$_4$ (final concentration 3 mM), DNaseI (final concentration 5 μg/mL), and EDTA (final concentration 10 mM), and then the pH was adjusted to 10.0 using NaOH. The remaining steps were conducted at 4°C. The cell lysate was subjected to centrifugation at 12,000 x g for 20 min to remove unlysed material. The supernatant was then subjected to ultracentrifugation at 100,000 x g for 30 min to pellet membrane

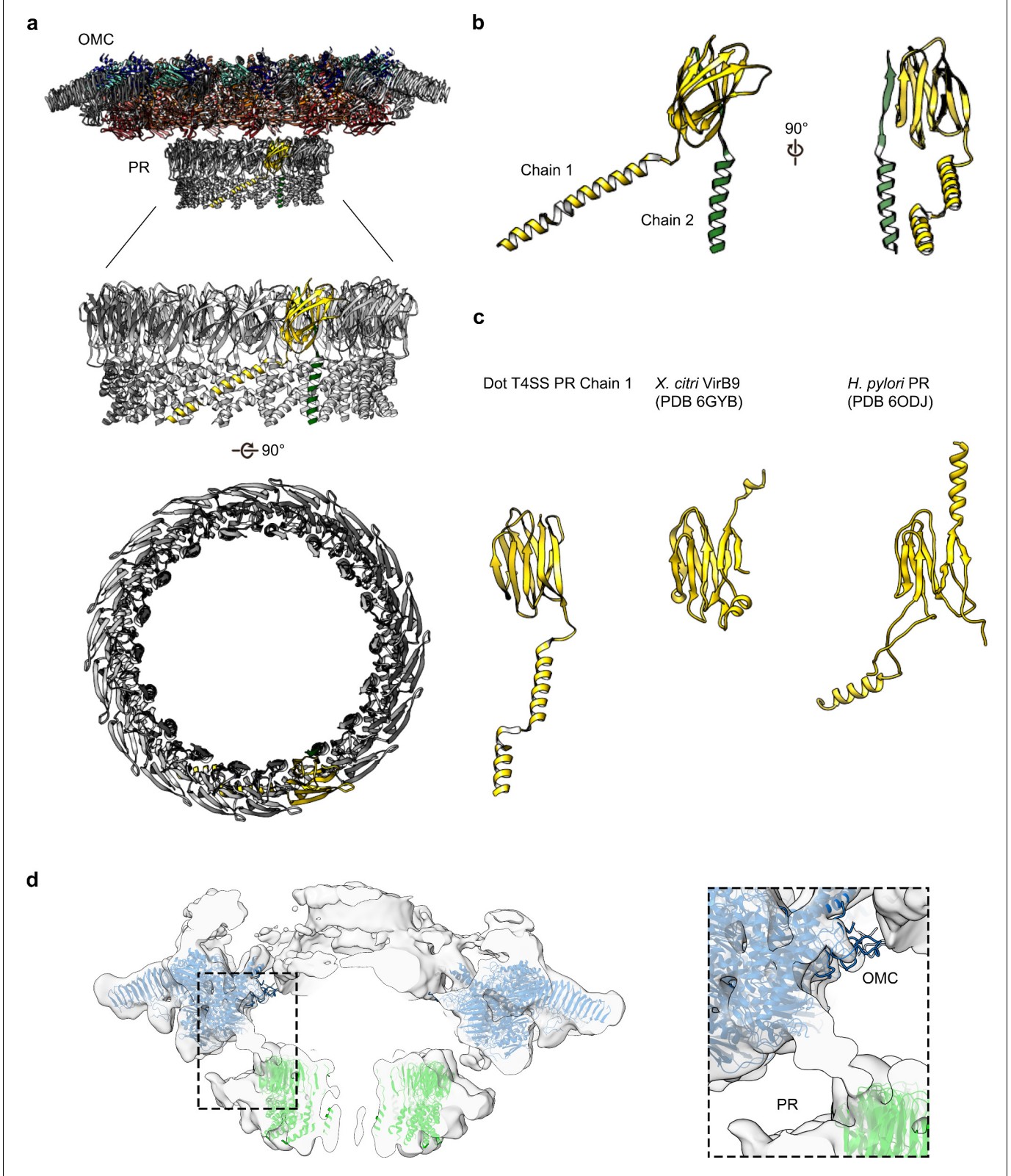

**Figure 9.** The PR of the Dot/Icm T4SS. (a) The Dot/Icm T4SS is comprised of two features known as the outer membrane cap (OMC) and the periplasmic ring (PR). Within the asymmetric unit of the PR we have modeled two chains (shown in yellow and green). (b) Chain 1 of the PR, shown in yellow, consists of two β-sheets and two α-helices. Chain 2, shown in green, consists of one β-strand and one α-helix. (c) Chain 1 of the Dot/Icm T4SS contains a globular fold that is similar to the N-terminus of VirB9 from *X. citri* and the core fold of the polyalanine model that was modeled in the PR of

*Figure 9 continued on next page*

*Figure 9 continued*

the Cag T4SS from *H. pylori*. (**d**) The physical connection between the OMC disk (blue) and PR (green) is shown with atomic models fit into the cryoEM reconstruction with no symmetry applied (gray).

complexes. The membrane complex pellets were resuspended and soaked overnight in a small volume of TET buffer (10 mM Trizma base pH 8.0, 1 mM EDTA, 0.1% Triton X-100). The resuspended sample was then subjected to centrifugation at 14,000 x g for 30 min to pellet debris. The supernatant was subjected to ultra-centrifugation at 100,000 x g for 30 min. The resulting pellet was resuspended in TET and complexes were further separated by Superose 6 10/300 column chromatography in TET buffer with 150 mM NaCl using an AKTA Pure system (GE Life Sciences). The sample collected from the column was used for microscopy and visualized by SDS-PAGE with silver staining (ProteoSilver Plus Silver Stain Kit). Mass spectrometry analysis was performed as described (*Anwar et al., 2018*).

### Cryo-EM data collection and map reconstruction – wild type T4SS

For cryo-EM, 4 µL of the isolated Dot/Icm T4SS sample was applied to a glow discharged ultrathin continuous carbon film on Quantifoil 2/2 200 mesh copper grids (Electron Microscopy Services). The sample was applied to the grid five consecutive times and incubated for ~60 s after each application. The grid was then rinsed in water to remove detergent before vitrification by plunge-freezing in a slurry of liquid ethane using a FEI vitrobot at 4°C and 100% humidity.

The images of the T4SS complexes from wild-type cells were collected on the Thermo Fisher 300 kV Titan Krios with Gatan K2 Summit Direct Electron Detector (DED) camera having a nominal pixel size of 1.64 Å. Micrographs were acquired using Leginon software (*Suloway et al., 2005*). The total exposure time was 16 s, and frames were recorded every 0.2 s, resulting in a total accumulated dose of 58.1 e$^-$ Å$^{-2}$ using a defocus range of −1 to −3 µm.

The video frames were first dose-weighted and aligned using Motioncor2 (*Zheng et al., 2017*). The contrast transfer function (CTF) values were determined using CTFFind4 (*Rohou and Grigorieff, 2015*). Image processing was carried out using cryoSPARC and RELION 3.0 (*Punjani et al., 2017*; *Bharat and Scheres, 2016*; *Zivanov et al., 2018*). Using the template picker in cryoSPARC, 771,806

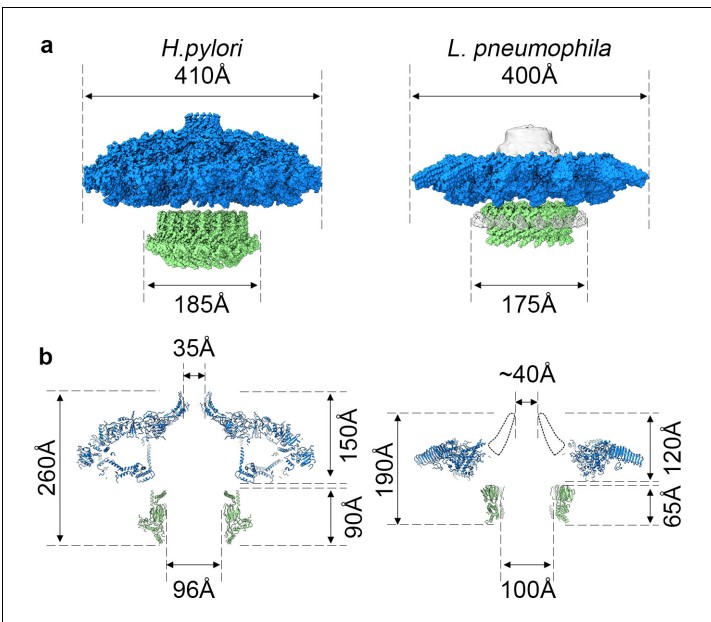

**Figure 10.** Comparison of *H. pylori* Cag T4SS and *L. pneumophila* Dot/Icm T4SS. (**a**) Surface view of atomic models for *Hp* Cag T4SS (left) and *Lp* Dot/Icm T4SS (right). OMC, blue; PR, green. For *Lp* Dot/Icm T4SS, C1 EM density is shown in gray for density not included in atomic models. (**b**) Secondary structure of *Hp* Cag T4SS (left) and *Lp* Dot/Icm T4SS (right). For *Lp* Dot/Icm T4SS, dome region outline is shown as a dotted line.

particles were picked from 3,594 micrographs and extracted using a 510 pixel box size (1.64 Å/ pixel). The extracted particles were used to generate representative 2D classes in cryoSPARC and approximately 20,800 particles were kept in good classes. These particles were then used for an *ab initio* model in cryoSPARC, which was then used as the reference for 3D auto-refinement with and without C13 symmetry (lowpass filtered to 40 Å). Finally, a solvent mask and B-factor were applied to improve the overall features and resolution of the 3D maps with and without C13 symmetry, resulting in reconstruction of 3D maps with a global resolution of 4.55 Å and 3.60 Å, respectively.

The C13 refined volume and corresponding particles were then exported to RELION for focused refinements. Estimation of beam-tilt values (CTF-refinement) was applied to the selected particles using RELION. With the CTF-refined particle stack, C13 symmetry-imposed refinement with a soft mask around the core complex was done, resulting in a 5.0 Å resolution 3D map.

For focused refinement of the OMC disk, signal subtraction for each particle containing the OMC disk was used with a soft mask. The subtracted particles were subjected to alignment-free focused 3D classification (three classes). The best 3D class of the OMC (~12,200 particles) was then subjected to a masked 3D refinement with local angular searches using C13 symmetry resulting in a 4.60 Å res- olution. Estimation of per-particle defocus values (CTF-refinement) was applied to the selected par- ticles using RELION. With the CTF-refined particle stack, C13 symmetry-imposed refinement with a soft mask around the OMC disk region of the Dot/Icm T4SS core complex was done, resulting in a 4.33 Å resolution 3D map that contained improved features. Post-processing resulted in the final OMC disk map with 3.5 Å resolution.

The same steps were followed for focused refinement of the PR, starting with signal subtraction for each particle containing the PR with a soft mask. The subtracted particles were subjected to alignment-free focused 3D classification (three classes). The best 3D class of the PR (~6850 particles) was selected based on class distribution (particle distribution), estimated resolution, and comparison of the 3D density maps. This class was then subjected to a masked 3D refinement with local angular searches using C18 symmetry resulting in a 7.54 Å resolution. Estimation of per-particle defocus val- ues (CTF-refinement) was applied to the selected particles using RELION. With the CTF-refined par- ticle stack, C18 symmetry-imposed refinement with a soft mask around the PR region of the Dot/Icm T4SS core complex was done, resulting in a 7.40 Å resolution 3D map that contained improved fea- tures. Post-processing resulted in the final PR map with 3.7 Å resolution. Map and model building data is summarized in *Appendix 1—table 1*.

## Cryo-EM data collection and map reconstruction – ΔDotG T4SS

For cryo-EM, 4 μL of the isolated ΔDotG T4SS sample was applied to a glow discharged Quantifoil 2/2 200 mesh copper grid with ultrathin (2 nm) continuous carbon film (Electron Microscopy Serv- ices). The sample was applied to the grid five consecutive times and incubated for ~60 s after each application. The grid was rinsed in water to remove detergent before vitrification by plunge-freezing in a slurry of liquid ethane using a FEI vitrobot at 22°C and 100% humidity.

The images of the T4SS complexes purified from the Δ*dotG* cells were collected by personnel at the National Center for CryoEM and Training (NCCAT) on the Thermo Fisher 300 kV Titan Krios with Gatan K2 Summit Direct Electron Detector (DED) camera having a nominal pixel size of 1.07 Å. Micrographs were acquired using Leginon software (*Suloway et al., 2005*). The total exposure time was 8 s and frames were recorded every 0.2 s, resulting in a total accumulated dose of ~65 e⁻ Å$^{-2}$ using a defocus range of −1.5 to −2.5 μm.

The video frames were first dose-weighted and aligned using Motioncor2 (*Zheng et al., 2017*). The CTF values were determined using CTFFind4 (*Rohou and Grigorieff, 2015*). Image processing was carried out using cryoSPARC and RELION 3.0 (*Punjani et al., 2017*; *Bharat and Scheres, 2016*; *Zivanov et al., 2018*). 120,367 particles were picked manually using Relion from 6990 micrographs and extracted using a 640 pixel box size (1.07 Å/pixel). The particles were imported into cryoSPARC, and the remaining processing steps were performed in cryoSPARC. The extracted particles were used to generate representative 2D classes and 9619 particles were kept in good classes. These par- ticles were used to generate two 3D *ab initio* models with C13 symmetry, the better of which con- tained 6342 particles. Homogeneous refinement of this model, also with C13 symmetry, resulted in a map with 4.2 Å resolution.

## Model building and refinement

A model was constructed from the OMC disk by first tracing all chains within the asymmetric unit using Coot (*Emsley et al., 2010*). We identified two folds that were similar to the crystal structure of DotD and thus docked the corresponding crystal structure (PDB 3ADY) into the map using UCSF Chimera (*Nakano et al., 2010*). All other chains were then iteratively built *de novo* in Coot and refined in PHENIX (*Afonine et al., 2018*). During subsequent rounds of model building and refinement it was noted that a second fold which was similar to DotK was present in the EM map which could not be identified as any of the known core components. The structure of DotK was then subjected to a protein fold analysis using the DALI server which returned Lpg0657 as a potential candidate (*Holm, 2019*). This crystal structure of Lpg0657 (PDB 3LDT) was then docked into the map using UCSF Chimera and the entire asymmetric unit was refined. The asymmetric unit was then duplicated in UCSF Chimera and each asymmetric unit docked into the map to generate a model of the entire OMC (*Pettersen et al., 2004*; *Rodríguez-Guerra Pedregal and Maréchal, 2018*). This structure was then refined in PHENIX with secondary structure and Ramachandran restraints applied (*Afonine et al., 2018*). During iterative rounds of refinement the nonbonded weighting parameter within PHENIX was optimized. A polyalanine model of the PR was constructed de novo in Coot and was refined using a similar protocol to that which was outlined above (*Emsley et al., 2010*).

To generate a model of the entire core complex, maps from the focused refinement of the OMC disk and PR were aligned in UCSF Chimera using the asymmetric reconstruction as a guide. The maps were then combined in PHENIX using phenix.combine_focus_maps with the models of the OMC disk and PR provided as additional templates (*Afonine et al., 2018*). The entire complex was then subjected to a round of refinement in PHENIX with secondary structure and Ramachandran restraints applied. *Figure 2—figure supplement 1* shows the Fourier shell correlations (FSCs) of the half maps against the refined model agree with each other, suggesting that the models are not over-refined.

To model the OMC reconstructed from the Δ*dotG* strain, the OMC disk from the wild type strain was docked into the map using UCSF Chimera (*Pettersen et al., 2004*). The model was then refined in PHENIX following a similar protocol to that which was outlined above (*Afonine et al., 2018*). Figures were made in part using ChimeraX (*Rodríguez-Guerra Pedregal and Maréchal, 2018*).

## Acknowledgements

The work presented here was supported by NIH R01AI118932 (MDO), F32 AI150027-01 (CLD), NIH 2T32DK007673 (MJS), S10OD020011 and the University of Michigan Department of Microbiology and Immunology (Swanson). Some of this work was performed at the National Center for CryoEM Access and Training (NCCAT) and the Simons Electron Microscopy Center located at the New York Structural Biology Center, supported by the NIH Common Fund Transformative High Resolution Cryo-Electron Microscopy program (U24 GM129539,) and by grants from the Simons Foundation (SF349247) and NY State. A portion of the molecular graphics and analyses was performed with UCSF Chimera, developed by the Resource for Biocomputing, Visualization, and Informatics at UC-San Francisco, with support from NIH P41-GM103311. Mass spectrometry experiments were performed by the University of Michigan Proteomics Resource Facility. The authors acknowledge Young-In (Eva) Kwon for assistance with figures. We thank the Cianfrocco, Cover, Lacy, and Ohi labs for helpful discussions. We acknowledge the use of the U-M LSI cryo-EM facility, managed by M Su, A Bondy, and L Koepping, and U-M LSI IT support. We thank U-M BSI and LSI for significant support of the cryo-EM facility.

## Additional information

### Funding

| Funder | Grant reference number | Author |
| --- | --- | --- |
| National Institute of Allergy and Infectious Diseases | F32 AI150027-01 | Clarissa L Durie |
| National Institute of General | S10OD020011 | Melanie D Ohi |

| Medical Sciences | | |
|---|---|---|
| National Institute of Allergy and Infectious Diseases | 2T32DK007673 | Michele Swanson |
| National Institute of Allergy and Infectious Diseases | R01AI118932 | Melanie D Ohi |

The funders had no role in study design, data collection and interpretation, or the decision to submit the work for publication.

## Author contributions

Clarissa L Durie, Conceptualization, Resources, Formal analysis, Supervision, Funding acquisition, Validation, Investigation, Visualization, Methodology, Writing - original draft, Project administration, Writing - review and editing; Michael J Sheedlo, Resources, Formal analysis, Supervision, Validation, Investigation, Visualization, Methodology, Writing - original draft, Project administration, Writing - review and editing; Jeong Min Chung, Data curation, Formal analysis, Validation, Investigation, Visualization, Methodology, Writing - original draft, Writing - review and editing; Brenda G Byrne, Resources, Data curation, Formal analysis, Validation, Investigation, Methodology, Writing - review and editing; Min Su, Conceptualization, Formal analysis, Validation, Investigation, Visualization, Methodology, Writing - original draft, Writing - review and editing; Thomas Knight, Resources, Project administration; Michele Swanson, Resources, Supervision, Methodology, Project administration, Writing - review and editing; D Borden Lacy, Resources, Formal analysis, Supervision, Validation, Investigation, Visualization, Methodology, Project administration, Writing - review and editing; Melanie D Ohi, Conceptualization, Resources, Formal analysis, Supervision, Funding acquisition, Investigation, Visualization, Methodology, Writing - original draft, Project administration, Writing - review and editing

## Author ORCIDs

Clarissa L Durie (iD) https://orcid.org/0000-0002-4027-4386
Michael J Sheedlo (iD) http://orcid.org/0000-0002-3185-1727
Jeong Min Chung (iD) http://orcid.org/0000-0002-4285-8764
Michele Swanson (iD) https://orcid.org/0000-0003-2542-0266
D Borden Lacy (iD) http://orcid.org/0000-0003-2273-8121
Melanie D Ohi (iD) https://orcid.org/0000-0003-1750-4793

## Decision letter and Author response

Decision letter https://doi.org/10.7554/eLife.59530.sa1
Author response https://doi.org/10.7554/eLife.59530.sa2

# Additional files

## Supplementary files

• Supplementary file 1. Proteins that copurify with Dot T4SS. a UniProtKB Accession Number b Proteins were identified by searching the MS/MS data against *L. pneumophila* (UniProt; 2930 entries) using Proteome Discoverer (v2.1, Thermo Scientific). Search parameters included MS1 mass tolerance of 10 ppm and fragment tolerance of 0.1 Da. False discovery rate (FDR) was determined using Percolator and proteins/peptides with a FDR of ≤1% were retained for further analysis. c Dot/Icm T4SS components are in bold.

• Supplementary file 2. Proteins that copurify with ΔDotG T4SS. [a]UniProtKB Accession Number [b]Proteins were identified by searching the MS/MS data against *L. pneumophila* (UniProt; 2930 entries) using Proteome Discoverer (v2.1, Thermo Scientific). Search parameters included MS1 mass tolerance of 10 ppm and fragment tolerance of 0.1 Da. False discovery rate (FDR) was determined using Percolator and proteins/peptides with a FDR of ≤1% were retained for further analysis.[c]Dot/Icm T4SS components are in bold.

• Transparent reporting form

## Data availability

All cryo-EM data included in this manuscript are available through the Electron Microscopy Data Bank (EMD-22068, EMD-22069, EMD-22070 and EMD-22071). All models that were constructed from these data are available via the Protein Data Bank (PDB 6x62, 6x64, 6x65, and 6x66).

The following datasets were generated:

| Author(s) | Year | Dataset title | Dataset URL | Database and Identifier |
|---|---|---|---|---|
| Durie CL, Sheedlo MJ, Chung JM, Byrne BG, Su M, Knight T, Swanson M, Lacy DB, Ohi MD | 2020 | Atomic model | http://www.rcsb.org/structure/6x62 | RCSB Protein Data Bank, 6x62 |
| Durie CL, Sheedlo MJ, Chung JM, Byrne BG, Su M, Knight T, Swanson M, Lacy DB, Ohi MD | 2020 | Atomic model | http://www.rcsb.org/structure/6x64 | RCSB Protein Data Bank, 6x64 |
| Durie CL, Sheedlo MJ, Chung JM, Byrne BG, Su M, Knight T, Swanson M, Lacy DB, Ohi MD | 2020 | Atomic model | http://www.rcsb.org/structure/6x65 | RCSB Protein Data Bank, 6x65 |
| Durie CL, Sheedlo MJ, Chung JM, Byrne BG, Su M, Knight T, Swanson M, Lacy DB, Ohi MD | 2020 | Atomic model | http://www.rcsb.org/structure/6x66 | RCSB Protein Data Bank, 6x66 |
| Durie CL, Sheedlo MJ, Chung JM, Byrne BG, Su M, Knight T, Swanson M, Lacy DB, Ohi MD | 2020 | Dot/Icm T4SS OMC WT | http://www.ebi.ac.uk/pdbe/entry/emdb/EMD-22068 | Electron Microscopy Data Bank, EMD-22068 |
| Durie CL, Sheedlo MJ, Chung JM, Byrne BG, Su M, Knight T, Swanson M, Lacy DB, Ohi MD | 2020 | Dot/Icm T4SS PR WT | http://www.ebi.ac.uk/pdbe/entry/emdb/EMD-22069 | Electron Microscopy Data Bank, EMD-22069 |
| Durie CL, Sheedlo MJ, Chung JM, Byrne BG, Su M, Knight T, Swanson M, Lacy DB, Ohi MD | 2020 | Dot/Icm T4SS | http://www.ebi.ac.uk/pdbe/entry/emdb/EMD-22070 | Electron Microscopy Data Bank, EMD-22070 |
| Durie CL, Sheedlo MJ, Chung JM, Byrne BG, Su M, Knight T, Swanson M, Lacy DB, Ohi MD | 2020 | Dot/Icm T4SS OMC ΔDotG | http://www.ebi.ac.uk/pdbe/entry/emdb/EMD-22071 | Electron Microscopy Data Bank, EMD-22071 |

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

# Appendix 1

**Appendix 1—table 1.** Map reconstruction and model refinement.

**Cryo-EM data collection and processing**

| | Dot/Icm T4SS OMC WT EMD-22068 | Dot/Icm T4SS PR WT EMD-22069 | Dot/Icm T4SS EMD-22070 | Dot/Icm T4SS OMC ΔDotG EMD-22071 |
|---|---|---|---|---|
| EMDB accession codes | | | | |
| **Data Collection and Processing** | | | | |
| Magnification | 18,000 | 18,000 | | 29,000 |
| Voltage (kV) | 300 | 300 | | 300 |
| Total Electron Dose (e-/Å2) | 58.1 | 58.1 | | ~65.0 |
| Defocus Range (μM) | −1 ~ −3 | −1 ~ −3 | | −1.5 ~ −2.5 |
| Pixel Size (Å) | 1.64 | 1.64 | | 1.07 |
| Processing Software | cryoSPARC/Relion | cryoSPARC /Relion | | cryoSPARC/Relion |
| Symmetry | C13 | C18 | | C13 |
| Initial Particles (number) | 771,806 | 771,806 | | 120,367 |
| Final Particles (number) | 12,200 | 6,850 | | 6,342 |
| Map Sharpening B Factor | −40.63 | −52.51 | | −55.3 |
| Map Resolution (Å) | 3.5 | 3.7 | | 4.2 |
| FSC Threshold | 0.143 | 0.143 | | 0.143 |
| **Model Refinement and Validation** | | | | |
| Refinement | | | | |
| Initial Model Used | 3ADY, 3LDT | N/A | N/A | N/A |
| Model Resolution | | | | |
| FSC (0.5) | 3.4 | 3.9 | 3.6 | 7.3 |
| FSC (0.143) | 3.2 | 3.6 | 3.2 | 4.2 |
| Model Composition (Residues) | 15,548 | 3,168 | 18,716 | 15,548 |
| DotC | 58–161, 173–268 | - | 58–161, 173–268 | 58–161, 173–268 |
| DotD$_1$ | 25–159 | - | 25-159 | 25-159 |
| DotD$_2$ | 2–161 | - | 2–161 | 2–161 |
| DotH | 273–361 | - | 273–361 | 273–361 |
| DotK | 41–180 | - | 41–180 | 41–180 |
| Lpg0657 | 99–234 | - | 99–234 | 99–234 |
| Unk OMC Chain 1 | 1-70 | - | 1-70 | 1-70 |
| Unk OMC Chain 2 | 1–127 | - | 1–127 | 1–127 |
| Unk OMC Chain 3 | 1-52 | - | 1-52 | 1-52 |
| Unknown PR | - | 1–32, 1–144 | 1–32, 1–144 | - |
| | | | | |
| Bond RMSD | | | | |
| Bond Length (Å) | 0.014 | 0.007 | 0.01 | 0.016 |
| Bond Angle (°) | 1.868 | 1.06 | 1.27 | 1.723 |
| **Validation** | | | | |
| Molprobity Score | 2.41 | 1.17 | 2.70 | 2.44 |
| Clashscore | 6.9 | 0.47 | 6.94 | 21.63 |

*Continued on next page*

| Ramachandran (%) | | | | |
|---|---|---|---|---|
| Favored | 91.5 | 91.5 | 92 | 87.7 |
| Allowed | 8.5 | 8.5 | 8 | 12.3 |
| Outliers | 0.0 | 0.0 | 0.0 | 0.0 |
| B Factor | 94.6 | 60.7 | 152.1 | 341.9 |
| PDB Accession Codes | 6 × 62 | 6 × 64 | 6 × 65 | 6 × 66 |

