## [Decision Letter]

**Acceptance summary:**

Type IV Secretion Systems (T4SS) are multi-component complexes used by bacteria to deliver effector proteins into eukaryotic cells during infection. This manuscript describes a high resolution cryo-EM structure of the T4SS from *Legionella pneumophila*, the causative agent of Legionnaires' Disease. This is a major advance for the T4SS field as the authors are able to definitively assign Dot proteins that make up the complex as well as their structures and details of their interactions at near atomic resolution.

**Decision letter after peer review:**

Thank you for submitting your article "Structural analysis of the *Legionella pneumophila* Dot/Icm Type IV Secretion System" for consideration by *eLife*. Your article has been reviewed by three peer reviewers, and the evaluation has been overseen by a Reviewing Editor and John Kuriyan as the Senior Editor. The following individual involved in review of your submission has agreed to reveal their identity: Peter Christie (Reviewer #3).

The reviewers have discussed the reviews with one another and the Reviewing Editor has drafted this decision to help you prepare a revised submission.

Summary:

This manuscript describes a high-resolution structure of the outer portion of the Dot/Icm Type IV secretion system (T4SS) elaborated by *Legionella pneumophila*. This portion of the T4SS is termed the core complex, and here the authors have solved the structure of the core complex by single-particle cryoelectron microscopy. This is a major advance for the T4SS field insofar as the authors were able to definitively assign Dot proteins that make up the complex as well as their structures and details of their interactions at near atomic resolution. The findings confirm a number of previous predictions derived from biochemical or lower resolution Cryoelectron tomography studies, and also identify new components with unexpected stoichiometries. Important advances are also made in defining the structures and symmetry mismatch of two sub-elements, here designated as the outer membrane cap (OMC) and periplasmic ring (PR). Overall, the findings allow for detailed comparisons between other large T4SSs such as the Helicobacter pylori Cag T4SS, which the authors solved last year, and other T4SSs, e.g., the pKM101 and Xanthomonas VirB systems, whose OMCs are less complex.

Essential revisions:

1) The title somewhat overstates the scope of the structural analysis presented. The title would be more informative/accurate to the reader if it clarified that the structural analysis was specific to the core complex rather than implicating the T4BSS as a whole.

2) Map and model quality analysis is absent. Please include FSC plots of maps versus models. Please include real-space correlation coefficient (CC) between map and model.

3) Results and Discussion, first paragraph. Is there any particular reason why the gel of purified sample is not shown and analysed at least in supplement, especially as Kubori and Nagai, 2019, is from a different group – it would complement the study. In Kubori and Nagai, 2019, the sample was gel filtrated on a Superose6. Is the complex used for this study also gel filtrated etc? And can you see the loss of DotG in the Δ KO strain on the gel?

---

## [Author Response]

Essential revisions:1) The title somewhat overstates the scope of the structural analysis presented. The title would be more informative/accurate to the reader if it clarified that the structural analysis was specific to the core complex rather than implicating the T4BSS as a whole.

The title has been revised to indicate that this study describes the core complex.

2) Map and model quality analysis is absent. Please include FSC plots of maps versus models. Please include real-space correlation coefficient (CC) between map and model.

The FSC plots of maps versus models are now included as Figure 2—figure supplement 1 (for WT) and Figure 8—figure supplement 2 (for mutant). As requested, we have included the real-space correlation coefficient (CC) as Figure 2—figure supplement 2 (for WT) and Figure 8—figure supplement 3 (for mutant).

3) Results and Discussion, first paragraph. Is there any particular reason why the gel of purified sample is not shown and analysed at least in supplement, especially as Kubori and Nagai, 2019, is from a different group – it would complement the study. In Kubori and Nagai, 2019, the sample was gel filtrated on a Superose6. Is the complex used for this study also gel filtrated etc? And can you see the loss of DotG in the δ KO strain on the gel?

The gels have been added to Figure 8—figure supplement 1. We observe DotG in the WT gel and not the gel from the mutant strain. The mutant complex was also purified using gel filtration on a Superose6. This and additional details about the strain used for this purification and complex isolation have been added to the Materials and methods under the subheadings “Preparation of Strains” and “Complex Isolation”.